# VESSA: Video-based objEct-centric Self-Supervised Adaptation for Visual Foundation Models

**Jesimon Barreto**[1]  **Carlos Caetano**[2]  **André Araujo**[3]  **William Robson Schwartz**[1]

[1]Departamento de Ciência da Computação, Universidade Federal de Minas Gerais (UFMG)
[2]Recod.ai, Instituto de Computação, Universidade Estadual de Campinas (UNICAMP)
[3]Google DeepMind

`{jesimonbarreto, william}@dcc.ufmg.br, caetanoc@unicamp.br, andrearaujo@google.com`

## Abstract

Foundation models have advanced computer vision by enabling strong performance across diverse tasks through large-scale pretraining and supervised fine-tuning. However, they may underperform in domains with distribution shifts and scarce labels, where supervised fine-tuning may be infeasible. While continued self-supervised learning for model adaptation is common for generative language models, this strategy has not proven effective for vision-centric encoder models. To address this challenge, we introduce a novel formulation of self-supervised fine-tuning for vision foundation models, where the model is adapted to a new domain without requiring annotations, leveraging only short multi-view object-centric videos. Our method is referred to as VESSA: **V**ideo-based obj**E**ct-centric **S**elf-**S**upervised **A**daptation for visual foundation models. VESSA's training technique is based on a self-distillation paradigm, where it is critical to carefully tune prediction heads and deploy parameter-efficient adaptation techniques – otherwise, the model may quickly forget its pretrained knowledge and reach a degraded state. VESSA benefits significantly from multi-view object observations sourced from different frames in an object-centric video, efficiently learning robustness to varied capture conditions, without the need of annotations. Through comprehensive experiments with 3 vision foundation models on 2 datasets, VESSA demonstrates consistent improvements in downstream classification tasks, compared to the base models and previous adaptation methods. Code is publicly available at https://github.com/jesimonbarreto/VESSA.

## 1   Introduction

Visual foundation models (VFMs) trained with self-supervised learning on large image datasets have become a powerful tool for a wide range of computer vision tasks [1, 2]. Techniques such as contrastive learning and self-distillation allow these models to learn high-quality visual representations without manual labels [3, 4]. Despite their generality, performance can suffer when applied to specialized domains with different characteristics from the pre-training data. For this reason, after the VFM is pre-trained, fine-tuning is commonly employed before applying it to downstream tasks. Supervised fine-tuning, in particular, has been the dominant approach, with impressive results across a variety of datasets and applications [2, 5] such as remote sensing [6, 7], medical imaging [8, 9] and place recognition [10, 11]. These successes demonstrate the adaptability of pre-trained models, but also highlight their reliance on labeled data, which can be expensive or impractical to obtain in many real-world scenarios.

39th Conference on Neural Information Processing Systems (NeurIPS 2025).

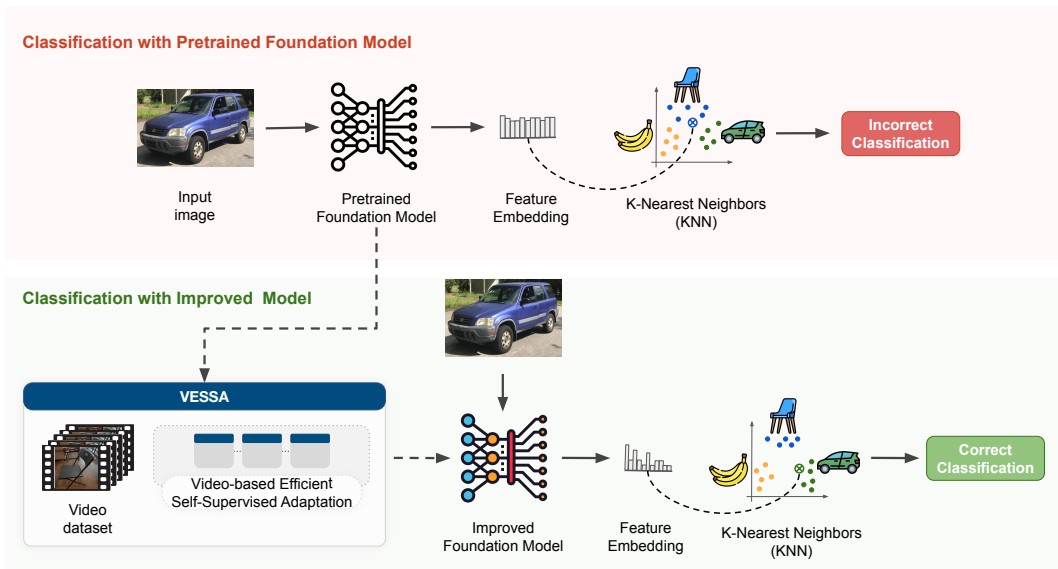

Figure 1: We present **VESSA**, a novel and efficient method for adapting vision foundation models using self-supervised fine-tuning with videos. Starting from a pretrained foundation model applied to a classification problem in a target domain, VESSA adapts the model without using labels by leveraging simple, object-centric videos. The resulting model learns improved representations that better structure the feature space in the target domain, boosting downstream classification accuracy.

Significant challenges may arise in scenarios where labeled data are unavailable for supervised fine-tuning. Despite its potential in cases where collecting annotations is costly, time-consuming, or even infeasible, unsupervised fine-tuning for foundation models remains largely underexplored in vision [12, 13]. By contrast, the natural language processing (NLP) community has long adopted unsupervised fine-tuning as a standard method to specialize large language models to new data distributions, typically via continued pretraining on unlabeled in-domain text [14, 15, 16]. While this strategy has proven successful for generative language models, its adaptation to visual data remains an open and challenging problem. For this reason, a few natural questions arise: how can we adapt a pre-trained vision model to a specific context without supervision? What forms of unlabeled visual data are best suited for adapting vision foundation models to new data distributions? What type of learning technique can effectively adapt pre-trained visual representations under the constraints in this scenario?

To answer the aforementioned questions, we propose VESSA (**V**ideo-based obj**E**ct-centric **S**elf-**S**upervised **A**daptation), a self-supervised fine-tuning method for VFMs that is both simple and effective, leveraging only short object-centric videos. A conceptual overview of the application of our model is presented in Fig. 1. VESSA employs a self-distillation training algorithm with critical adaptations to make it work in a fine-tuning setup. We show that a naive application of self-distillation to the fine-tuning stage may lead to a degraded model state, but this can be avoided with the careful adjustments proposed in this work. In particular, we introduce a training schedule which adjusts the self-distillation prediction head before unfreezing the rest of the model. Then, an efficient method is used to gently tune the backbone parameters towards the new domain without disrupting the encoded pre-trained knowledge, also leveraging uncertainty weighting to prioritize harder training examples. Finally, we propose to source observations of target objects in the new domain from short videos, which are easy to capture and require no labeling, but enhance the model performance significantly.

Experimentally, we leverage three existing foundation models and two downstream classification applications to comprehensively assess the proposed VESSA technique. Our results demonstrate that the proposed video-based self-supervised fine-tuning significantly outperforms base foundation models or other fine-tuning strategies.

## 2 Related Work

Recent advances in visual foundation models have reshaped the landscape of computer vision by enabling scalable, general-purpose representations trained on massive datasets. To tailor these representations to specific downstream tasks, task-adaptive fine-tuning strategies have emerged as a solution for many applications, aiming to bridge the gap between foundation model generality and task-specific performance. Complementary to this, approaches in video-to-image knowledge transfer explore how temporal and multimodal supervision in video models can be distilled into stronger static image representations. In this section, we provide a structural overview of these areas, highlighting their connections to our proposed formulation and identifying key gaps our method addresses.

**Self-supervised Visual Foundation Models.** Self-supervised transformer-based foundation models have achieved state-of-the-art results across a wide range of computer vision tasks [17, 18, 19]. For image classification, image-level self-supervised models such as DINO [3], DINOv2 [4], iBOT [18], and SimCLR [19] have shown strong performance, with DINO and DINOv2 notably relying on label-free self-distillation for scalable pretraining. The widely adopted Masked Autoencoders (MAE) [17] build on the idea of reconstructive pretraining by randomly masking a large portion of input patches and training the model to reconstruct them, enabling efficient learning of high-capacity visual representations that scale well with data and model size. A key advantage shared by all these methods is their independence from human annotations, which allows them to leverage large-scale unlabeled datasets without the constraints and costs associated with manual labeling. This characteristic makes them especially powerful in scenarios where labeled data is scarce or unavailable, enabling the use of virtually all accessible visual data for representation learning.

**Task-Adaptive Fine-Tuning.** Task-adaptive or continual pretraining has yielded notable gains in natural language processing, enabling foundation models to specialize for downstream domains [14, 15, 16]. In computer vision, the use of self-supervised learning to adapt models to different domains has been widely studied [20, 21, 22, 23], where the goal is to bridge the gap between a labeled source domain and an unlabeled target domain. In contrast, we aim to adapt a pretrained VFM to a specific domain using only unlabeled data. Many approaches for adapting vision foundation models have been proposed, such as AdaptFormer [24] and Visual Prompt Tuning [25], but they rely on supervised learning and often involve more complex adaptation pipelines. Recent works such as [26, 27] employ continual pretraining to adapt image-based foundation models to satellite imagery. These approaches aim to construct new domain-specific foundation models, which are subsequently fine-tuned with supervision for downstream tasks. ExPLoRA [28], the most similar to our approach in terms of weight adaptation, builds a new foundation model for satellite images by adapting self-supervised models trained on other domains—such as DINOv2 [4] or Masked Autoencoders [17]—and incorporates LoRA [29] for efficient continual self-supervised learning. It reuses the training procedures and heads of the base models, but ultimately still relies on supervised fine-tuning to complete the adaptation. In contrast, our method does not propose a new foundation model. We reuse the original DINO architecture [3] and perform direct adaptation to downstream tasks without requiring labeled data. We adapt it not only with a new loss function, but also with a new training method, with careful tuning of different parts and the integration of an efficient parameter learning component. This enables effective representation learning even in settings with limited domain-specific data.

**Video to Image Knowledge Transfer.** Videos provide rich supervisory signals due to their inherent spatio-temporal consistency, natural motion, and realistic object transformations [30, 31, 32, 33, 34, 35]. Recently, several works have investigated transferring knowledge from video to image representations [36]. Methods such as VITO [37] and ViC-MAE [38] build joint models for video and image tasks, involving costly frame selection pipelines and hybrid loss formulations (e.g., masking and contrastive objectives). These models often require substantial architectural modifications and exhibit training instability, with heavy reliance on masking to guide object learning. Time Does Tell [39] introduces a dense frame-wise module to extract visual features from all frames. This highlights the potential of video data and its valuable contribution to unsupervised representation learning. In contrast, we propose a lightweight approach using object-centric videos with minimal visual clutter. Our method emphasizes learning unified and robust representations from simple videos, enabling improved generalization without extensive frame curation or resource consumption. Importantly, most prior video-based methods target pixel-level tasks (e.g., segmentation, detection) and show

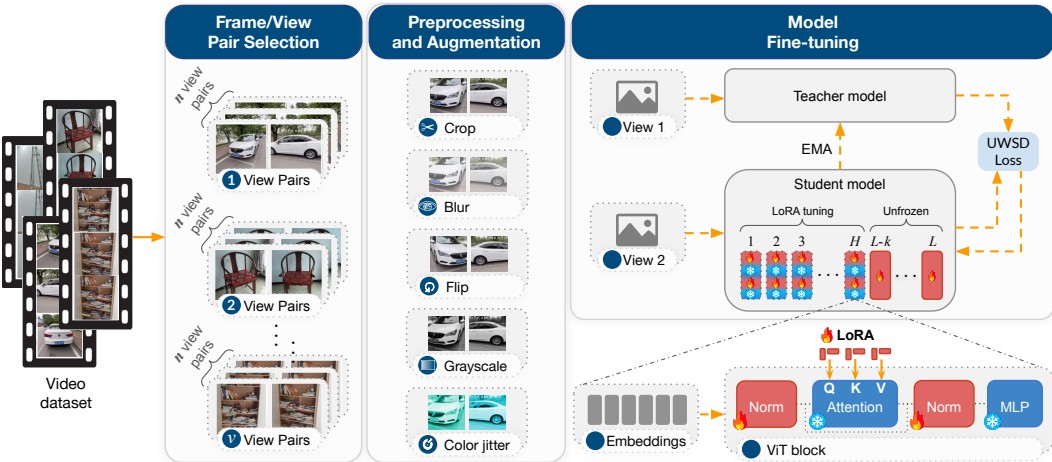

Figure 2: **The proposed training pipeline.** The model input consists of videos, which first undergo a *Frame Selection* stage, where $n$ pairs of frames are sampled from each video. These pairs are then passed through the *Preprocessing and Augmentation* stage, where distinct transformations are applied to the first and second images of each pair. The resulting views are fed into teacher and student networks, both initialized from the same foundation model; LoRA is applied to their architectures for parameter-efficient fine-tuning. Finally, the *uncertainty-weighted self-distillation loss* (UWSD) is applied to align their representations.

limited improvements for frame-level classification. These methods also typically require fine-tuning to be competitive.

## 3 VESSA

In this section, we describe the methodological foundations of our work and introduce our novel formulation of self-supervised fine-tuning for vision foundation models. We begin by revisiting prior methods that serve as the basis for our formulation, highlighting their key mechanisms. Building on this foundation, we then present our novel self-supervised fine-tuning strategy for vision foundation models, designed to enhance adaptation.

### 3.1 Background

Our method builds on recent advances in self-supervised learning and parameter-efficient adaptation. We focus on two core components: DINO [3], a self-distillation framework for label-free representation learning, and LoRA [29], a lightweight technique for adapting large models with minimal trainable parameters. We review them briefly in the following.

**DINO** [3] is a self-supervised learning framework that trains a student network to match the output of a teacher network, with both networks receiving different augmented views of the same image. The teacher's parameters are updated using an exponential moving average (EMA) of the student's weights, ensuring stable training and consistent representations. The method aligns the output probability distributions of the student and teacher using a cross-entropy loss:

$$\mathcal{L}_{\text{DINO}} = -\sum_i f_t(x_{t,i}) \log f_s(x_{s,i}) \tag{1}$$

where $x_{t,i}$ and $x_{s,i}$ denote the augmented views for the $i$-th image, and $f_t(\cdot)$ and $f_s(\cdot)$ represent the outputs of the teacher and student networks, respectively. This objective minimizes the discrepancy between their normalized output distributions, encouraging the student network to learn meaningful representations without the need for labeled data. DINO has demonstrated strong performance across a variety of visual tasks, producing robust and transferable features.

**Low-Rank Adaptation (LoRA)** [29] is a parameter-efficient fine-tuning technique that enables the adaptation of pre-trained models by injecting trainable low-rank matrices into their weight structure. Rather than updating the full weight matrix $W \in \mathbb{R}^{d \times k}$ of a linear layer during training, LoRA keeps $W$ frozen and instead learns an additive update in the form of a low-rank decomposition:

$$\Delta W = AB, \quad A \in \mathbb{R}^{d \times r}, \quad B \in \mathbb{R}^{r \times k}, \tag{2}$$

where $r \ll \min(d, k)$. Here, $A$ and $B$ are the newly introduced trainable matrices, and $\Delta W$ is the low-rank adaptation added to the original weight matrix. This approach significantly reduces the number of trainable parameters and memory requirements while preserving model performance, making it particularly effective for adapting large-scale models in resource-constrained environments.

### 3.2 Video-based objEct-centric Self-Supervised Adaptation (VESSA)

The VESSA pipeline is illustrated in Figure 2, and consists of three main modules: *Frame Selection*, *Preprocessing and Augmentation*, and *Model Fine-tuning*. Each input video $V$, with $T$ frames $\{F_i\}_{i=1}^{T}$ is first processed by the **Frame Selection** module, which samples $n$ frame pairs per video. For each pair of frames that we aim to construct, we first randomly sample a starting frame index $t \sim \mathcal{U}(1, T - \delta_{\max})$, where $\delta_{\max}$ is a predefined maximum temporal offset, and $\mathcal{U}$ denotes the uniform distribution. Then, we sample frames based on a temporal gap $\delta \sim \mathcal{U}(1, \delta_{\max})$, ensuring a minimum offset of one frame. Each selected pair of frames is then formed according to the rule:

$$\delta \in [1, \delta_{\max}], \quad t \in [1, T - \delta]$$

This randomized strategy introduces temporal diversity by allowing variable distances between frames, which helps the model learn more robust representations across different viewpoints. At the end of this module, we obtain a batch composed of frame pairs sampled from different videos, represented as:

$$\mathcal{B} = \left\{ (F_{t_k}, F_{t_k + \delta_k}) \mid k = 1, \ldots, v \right\},$$

where each $(F_{t_k}, F_{t_k + \delta_k})$ is a pair of frames from the $k$-th video in the batch, and $v$ is the batch size. The temporal index $t_k$ and offset $\delta_k$ are sampled independently for each pair. This setup ensures that the batch contains diverse video content and temporal gaps.

In the **Preprocessing and Augmentation** module, each frame in the pair $k$ is transformed independently using two distinct pipelines of random augmentations. This results in two augmented views: $F_t^{(a)}$ and $F_{t+\delta}^{(b)}$, which are designed to promote appearance diversity and representation robustness. This module also generates the *local crops* used in our adaptation. Inspired by the reference method, which employs multiple local crops per image to stabilize fine-tuning, we adapt this strategy by sampling local crops as pairs—one from each frame. This pairing ensures temporal consistency while preserving local variability, contributing to more robust and transferable representations. At the end of this module, we have

$$\mathcal{B} = \left\{ \left( F_{t_k}^{(a)}, F_{t_k + \delta_k}^{(b)}, \left\{ F_{t_k}^{(c_i)}, F_{t_k + \delta_k}^{(c_i)} \right\}_{i=1}^{u} \right) \mid k = 1, \ldots, v \right\},$$

where:

- $F_{t_k}^{(a)}$ and $F_{t_k + \delta_k}^{(b)}$: two temporally separated frames from the $k$-th video, with distinct global transformations $a$ and $b$ applied;

- $\left\{ F_{t_k}^{(c_i)}, F_{t_k + \delta_k}^{(c_i)} \right\}_{i=1}^{u}$: a set of $u$ pairs of local crops, where $c_i$ denotes a distinct transformation involving a small crop on the main frame; and $u$ the number of local crop pairs.

In the **Model Fine-tuning** stage, training is performed in batches; however, for clarity, we describe the process using a single pair of images and associated local crops. The pair $k$, given by $(F_t^{(a)}, F_{t+\delta}^{(b)})$, is passed through both the student and teacher networks, while the local crops are processed only by the student. Both networks are initialized from the same pretrained vision foundation model. The outputs of the student and teacher are denoted as follows:

$$s = f_{\text{s}}(F_t^{(a)}), \quad q = f_{\text{t}}(F_{t+\delta}^{(b)}), \quad s_{lc1} = f_{\text{s}}(F_t^{(c)}), \quad s_{lc2} = f_{\text{s}}(F_{t+\delta}^{(c)})$$

$s, q$ denote the outputs of the projection head applied to the global frame representations. The final representations of all local crops (i.e., $s_{lc1}$ and $s_{lc2}$), along with $s$, are also compared to $q$ as part of the DINO loss computation. The fine-tuning training objective is a weighted form of the formulation presented in Section 3.1. To prioritize uncertain teacher outputs, we introduce an *Uncertainty-Weighted Self-Distillation (UWSD)* loss, which modulates the contribution of each sample to the loss based on the estimated uncertainty of the teacher's predictions. The entropy of the teacher's distribution is computed. The entropy is used to compute the weight:

$$w(q) = 1 + \gamma \cdot \mathcal{H}(q),$$

where $\gamma$ is a hyperparameter controlling the influence of uncertainty. The final training objective becomes:

$$\mathcal{L}_{\text{UWSD}} = \frac{1}{N} \sum_{(q,s,s_{lc_i}) \in \mathcal{B}} w(q) \cdot \mathcal{L}_{\text{DINO}}(q, s, s_{lc_i})$$

**Critical optimization considerations.** Continuing self-supervised training from a pretrained foundation model requires careful handling due to potential gradient instabilities. These are often caused by shifts in data distribution and the use of a randomly initialized projection head, which introduces abrupt gradient changes early in training. In standard training regimes, all network parameters are typically updated simultaneously, which can lead to unbalanced gradient flow and degrade the pretrained representations. To mitigate this, we initially freeze the backbone and train only the projection head for a few epochs, allowing it to adapt to the existing embedding space. As another strategy to mitigate this issue, we gradually unfreeze the backbone, applying different strategies to different parts of the network. Specifically, we enable fine-tuning of the first $H$ layers using LoRA [29], which restricts updates to low-rank adaptations of the attention weights—specifically in the Query, Key, and Value projections of each self-attention layer—while keeping the normalization layers trainable. This design helps preserve the low-level visual features encoded in early layers, such as edges and textures, which are generally transferable across domains. In contrast, the last $L$ layers of the backbone are fully unfrozen and updated normally, allowing the model to adapt high-level semantic representations to the new domain. This staged unfreezing strategy mitigates representational drift while preserving stability and efficiency during fine-tuning.

## 4 Experiments

### 4.1 Experimental Setup

**Datasets.** MVImageNet[40] and CO3D [41] are large-scale video datasets offering multi-view images. MVImageNet comprises 6.5 million frames from over 219,000 videos across 238 object categories, while CO3D includes 1.5 million frames from 19,000 videos spanning 50 categories. Captured from various viewpoints under real-world conditions, these datasets enable learning of viewpoint-consistent representations. To adapt them for classification, we designed a protocol that splits each class into training and testing sets (75%-25%), selecting one frame per video and using $k$-Nearest Neighbors (KNN) to evaluate the quality of learned embeddings across different views and instances.

**Implementation details.** Our experiments were performed on TPU v3-8, featuring 8 cores and 128 GB of high-bandwidth memory. All implementations used the `scenic` library [42] in JAX. We adopted the ViT-Base architecture to balance performance and efficiency, as preliminary tests showed that the Small model underfit and the Large model offered minimal gains at higher cost. Our training followed the base hyperparameter configuration of the DINO protocol [3], except for the specific settings detailed below. As a reference, we adopted 10 training epochs for both the initial projection head adaptation and the subsequent full model training, using a batch size of 256 and an input image resolution of 224×224. For each video, we sampled 3 frame pairs. The hyperparameter $\gamma$, which controls the weight of the distillation loss, was set to 1. For the image-based baseline, we used the

first frame of each pair as the reference image. This frame was then processed through the subsequent steps as a single-image input, following the standard image-based pipeline.

**Statistical significance test.** To assess the significance of the observed differences, we performed statistical tests using three independent runs for each experimental configuration. We employed an unpaired Student's $t$-test with a 90% confidence level. In supplementary material we report confidence intervals for the main comparisons to highlight cases where the differences were statistically significant.

**Visual Foundation Models** Our experiments were conducted using widely adopted and state-of-the-art VFMs, including DINO [3], DINOv2 [4], and TIPS [43]. These models represent a strong set of visual backbones commonly used in computer vision tasks. In order to standardize the experiments, an input size of 224 was used for all VFMs. However, please note that TIPS can achieve improved results with an input size of 448, which would match its pretraining setup.

We compare our method with ExPLoRA [28], a recent approach for improving transfer learning of pretrained vision transformers (ViTs) under domain shifts. To ensure a fair comparison with our video-based approach, we extend ExPLoRA to operate on short object-centric videos. Experiments using TIPS + ExPLoRA were not reported since the original work ExPLoRA clearly specifies that it is designed for continual self-supervised learning using training heads from self-supervised models (e.g., the projection head from DINO for self-distillation or autoencoder-based modeling such as MAE [44]). TIPS, however, is not a self-supervised model.

## 4.2 Results

We begin our evaluation by analyzing the individual contributions of each component of the VESSA approach through a comprehensive ablation study. This analysis provides insights into the effectiveness of leveraging video data for self-supervised adaptation and highlights the critical design choices that enable our approach to outperform existing alternatives. In what follows, we systematically isolate and compare different configurations, followed by comparisons to state-of-the-art baselines across multiple datasets and model architectures.

A series of experiments conducted on the MVImageNet dataset using the vision transformer-small (ViT-S) to isolate and evaluate the contributions of each component of our method are shown in Table 1. Given the challenge of adapting to a new domain with limited and unlabeled data, we first trained DINO from scratch using both image and video data. As expected, the results were suboptimal due to the limited data, with a final accuracy of 33.86%. However, training with video (i.e., using different views of the object from the video) consistently outperformed the image-based counterpart, achieving 39.39% accuracy—an improvement of 5.53 percentage points (p.p.) aligning with our motivation to leverage temporal information—though the results remain relatively low overall. Subsequently, we applied the pretrained DINO model directly, which yielded solid performance and served as a strong baseline, achieving 89.69% accuracy. We also evaluated a naive continuation of training using only images, which similarly led to performance degradation, resulting in a slightly lower accuracy of 88.54%. Notably, switching from image input with local crops (88.54%) to video input without local crops (90.53%) already provided a larger gain (1.99 p.p.), indicating that temporal information plays a stronger role than local crops alone. Careful decision of the training head projection before fine-tuning had a significant impact, improving performance by approximately 10 p.p.. For instance, using video input with local crops but without training the head achieved 80.87%, whereas enabling head training in the same setup increased accuracy to 91.87%, underscoring that head training is the dominant factor. This result indicates the importance of carefully designed adaptations to achieve such improvements. Finally, we compared our full method against its individual variants, confirming that our complete approach achieves the best results, validating the importance of each design choice in the overall effectiveness of VESSA. The best-performing configuration achieved 91.87% accuracy, representing an improvement of 2.18 p.p. over the pretrained DINO baseline, with the optimal setting obtained by unfreezing the last 2 layers during adaptation.

Across all experiments, leveraging video consistently outperforms frame-based alternatives. To better understand the source of these gains—whether from motion cues or temporal continuity—we conducted an additional experiment on the CO3D dataset using DINO and DINOv2. Specifically, we evaluated the impact of frame distance ($\delta$) during self-supervised fine-tuning, as detailed in Table 2. The results show that varying the temporal gap between frames affects performance, with the highest

Table 1: **Ablation study on components for video-based self-supervised ViT fine-tuning.** All models use ViT-S/16 and are evaluated with $k$-NN ($k=1$) on the MVImageNet dataset. We analyze the impact of architectural choices, local crops, training heads, and data modalities.

| Method | UWSD Loss | Unfrozen Last Layers | Local Crops | Train Head | Input | Accuracy (%) |
|---|---|---|---|---|---|---|
| | ✓ | 2 | ✓ | ✓ | Video | **91.87** |
| | ✓ | 2 | | ✓ | Video | 90.53 |
| | | 2 | ✓ | ✓ | Video | 90.92 |
| VESSA | ✓ | 1 | ✓ | ✓ | Video | 87.14 |
| | ✓ | 3 | ✓ | ✓ | Video | 90.80 |
| | ✓ | 4 | ✓ | ✓ | Video | 90.55 |
| | ✓ | 2 | ✓ | | Video | 80.87 |
| | ✓ | 2 | ✓ | ✓ | Image | 88.54 |
| DINO [3] | | | | ✓ | Image | 33.86 |
| DINO [3] | | | | ✓ | Video | 39.39 |
| DINO [3] Pretrained | | | | | Image | 89.69 |

accuracy achieved when $\delta$ was randomly sampled from the range [5, 10], yielding 85.03% with DINO and 91.85% with DINOv2. This suggests that exposing the model to diverse temporal relationships between frames contributes positively to representation learning.

Table 2: **Top-1 accuracy (%) on the CO3D dataset using different frame distance strategies.** We report k-Nearest Neighbors (k=1) classification accuracy for models pretrained with DINO and DINOv2. Each value of $\delta$ defines the temporal distance between frames selected from videos during self-supervised fine-tuning.

| Frame Distance ($\delta$) | DINO [3] | DINOv2 [4] |
|---|---|---|
| 1 | 85.00 | 91.51 |
| 2 | 84.73 | 91.52 |
| 3 | 84.90 | 91.39 |
| 4 | 84.27 | 91.42 |
| 5 | 84.66 | 91.80 |
| 10 | 85.00 | 91.74 |
| 15 | 84.54 | 91.25 |
| 20 | 84.82 | 91.23 |
| Random [5, 10] | **85.03** | **91.85** |
| Random [10, 30] | 82.46 | 91.52 |

After analyzing the individual components of our approach, we now turn to a broader evaluation of VESSA applied to different backbone models across two datasets. As shown in Table 3, all other methods significantly outperform the base pretrained models without fine-tuning (the first row of the tables), except for a few cases involving a baseline variant of our method that employs static images only, which we refer to as Static-baseline. In particular, when video data is used, unsupervised fine-tuning generally leads to superior performance across both datasets and architectures, with the exception of Explora with video and DINO in MVImgNet, where performance decreases relative to the pretrained model. The performance gap between VESSA and Static-baseline, as well as between the ExPLoRA baseline and ExPLoRA + video, confirms the effectiveness of adapting models to the target domain using unlabeled video data in both CO3D and MVImageNet datasets. In contrast, the results of the Static-baseline indicate that a naive image-based self-supervised continual learning approach is not sufficient to achieve successful fine-tuning.

Considering the CO3D dataset, applying VESSA to DINOv2 yields the best result of 91.85% $\pm$ 0.56, which is 2.21 p.p. higher than ExPLoRA + video (89.64 $\pm$ 0.47); this difference is statistically significant. On the other hand, for the MVImageNet dataset, VESSA achieved 96.01 $\pm$ 1.08 while ExPLoRA + video reached 96.15 $\pm$ 0.87; however, the difference is not statistically significant (see supplementary material for details). Nevertheless, when considering DINO, VESSA outperforms again the ExPLoRA results. Moreover, our approach also performs better with TIPS against its

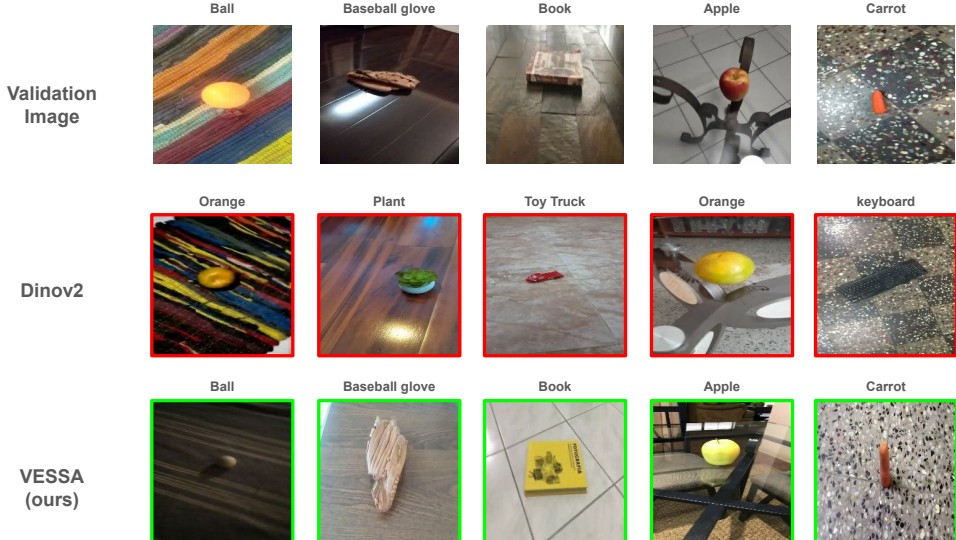

Figure 3: Qualitative examples of nearest neighbor retrieval ($k = 1$) on the CO3D validation set. We selected some of the most challenging validation samples and retrieved the nearest neighbors using two methods, shown row-wise: the first row displays the query (validation) images; the second row presents the retrievals using raw DINOv2 features; the third row shows the retrievals produced by our method. Images with red borders indicate incorrect retrievals, while green-bordered images represent correct ones. These examples illustrate the effectiveness of leveraging multi-frame video information for representation learning. Notably, our method demonstrates greater focus on the object of interest, whereas the baseline often retrieves matches dominated by background similarity.

pretrained version. It is important to present these results—alongside those in Table 1—as they demonstrate that simply continuing self-supervised training with domain-specific image data, while seemingly straightforward, does not yield consistent improvements. As the results show, this strategy often leads to performance degradation or, at best, no noticeable gains.

Table 3: **Top-1 accuracy (%) on CO3D and MVImageNet datasets using k-Nearest Neighbors (k=1).** We compare pretrained vision foundation models, an image-based baseline, and our proposed video-based fine-tuning method. All results are reported on the validation set using representations extracted from the backbone and evaluated via KNN. Our method achieves superior performance by leveraging object-centric videos for unsupervised adaptation.

| Method | CO3D | | | MVImageNet | | |
| | DINO [3] | DINOv2 [4] | TIPS [43] | DINO [3] | DINOv2 [4] | TIPS [43] |
|---|---|---|---|---|---|---|
| Pretrained | 78.86 | 87.86 | 60.02 | 90.44 | 95.75 | 78.65 |
| ExPLoRA [28] | 79.78 | 88.31 | — | 90.94 | 95.79 | — |
| ExPLoRA [28]+video | 83.64 | 89.64 | — | 87.74 | **96.15** | — |
| Static-baseline | 80.31 | 81.60 | 55.59 | 89.39 | 92.53 | 76.05 |
| VESSA | **85.03** | **91.85** | **70.56** | **92.51** | 96.01 | **80.54** |

To qualitatively illustrate the benefits of video-based self-supervised training, Figure 3 shows examples of top retrievals using KNN based on the learned embeddings. When comparing the pretrained DINOv2 model with our proposed VESSA method, we observe that DINOv2 produces embeddings that focus primarily on the background and broad scene structures. In contrast, VESSA clearly attends to the object of interest, even in challenging cases where the texture or color of the retrieved object differs from the query image. This demonstrates that VESSA learns more semantically meaningful and object-centric representations, which improves robustness and task relevance.

# 5   Conclusions

In this work, we presented  VESSA  (**V**ideo-based obj**E**ct-centric **S**elf-**S**upervised **A**daptation), a simple and effective strategy for unsupervised fine-tuning of visual foundation models using object-centric videos. Our approach requires no labeled data and leverages temporal coherence by treating distinct frames from the same video as positive pairs in a contrastive setup. Inspired by advances in NLP, we showed that unsupervised fine-tuning in vision is both feasible and valuable. We demonstrated that our method consistently improves classification performance on domain-specific datasets while remaining lightweight and easy to apply. This opens up new directions for adapting foundation models to new visual contexts without additional supervision, making them more versatile and accessible in practice.

**Limitations** A notable limitation of our approach is the tendency to forget previously acquired knowledge during fine-tuning —a known drawback of fine-tuning methods in general. Additionally, our experimental setup relies on video data that offer multiple viewpoints of the same object, a characteristic that is not commonly available in many real-world datasets.  This may limit the applicability of the approach to scenarios where such structured multi-view data is not present.

## Acknowledgments and Disclosure of Funding

The authors would like to thank the National Council for Scientific and Technological Development – CNPq (Grant 312565/2023-2), Fundação de Amparo à Pesquisa do Estado de São Paulo – FAPESP (Grant 2023/12086-9), Recod.ai and Google LLC.

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

# A   Technical Appendices and Supplementary Material

In this Technical Appendix and Supplementary Material, we present the complementary experiments of the paper. These evaluations include the main experiments with confidence intervals for the primary results, demonstrating the statistical significance of the observed differences. We further analyze the input differences between DINO and VESSA, and investigate the impact of applying stronger image augmentations, aiming to reduce the gap between static images and the visual variability observed in video data. We also present an additional study examining the extent of catastrophic forgetting when adapting visual foundation models with VESSA, highlighting its impact on general-purpose performance. Finally, we provide a detailed analysis of the training cost associated with VESSA, offering quantitative insights into its computational efficiency and practical feasibility.

**The main results with confidence intervals** are shown in Tables 4 and 5, which report the top-performing models along with confidence intervals to highlight the significance of the performance differences. The results suggest that VESSA achieves significantly better performance in several cases. We employed an unpaired Student's $t$-test with a 90% confidence level. We report confidence intervals for the main comparisons to highlight cases where the differences were statistically significant. For the CO3D dataset, the variation in accuracy across runs was $0.52$ for DINO, $0.56$ for DINOv2, and $1.03$ for TIPS. In all cases, the differences were statistically significant when compared to the second-best performing method, which in this case was ExPLoRA—a method also based on video data. In contrast, on the MVImageNet dataset, the variations were $1.11$ for DINO, $1.08$ for DINOv2, and $1.71$ for TIPS. In this scenario, non-overlapping confidence intervals between the DINO-based baseline and VESSA (ours) indicate a statistically significant difference. In the case of DINOv2 pretrained on MVImageNet, a noticeable overlap between the two video-based methods can be observed.


VESSA (ours)      DINO/DINOv2


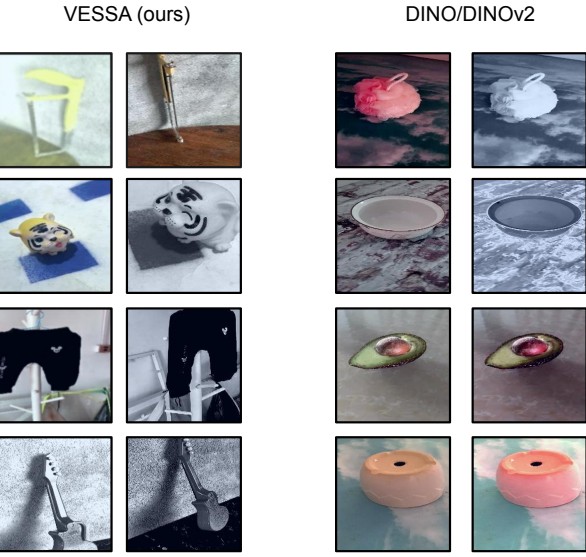

Figure 4: Example frames from the MVImageNet dataset illustrating the differences between global crop input pairs used for the teacher and student networks during training with DINO and VESSA (ours). Our method, VESSA, introduces substantially greater variability in the appearance of the evaluated object. The temporal distance between the selected frames is $\delta = 5$ frames. The first image of each pair shows the global crop from the transformation of view 1, and the second image of each pair shows the global crop corresponding to the transformations of view 2.

**Approximate video-like image transformations.** Motivated by the strong performance observed when training with videos, we investigated whether additional image transformations—beyond those used in the standard DINO pipeline—could simulate the benefits of camera motion. To this end, we applied a set of motion-inspired augmentations to one of the views during training, aiming to mimic the effect of slight viewpoint changes. Specifically, we incorporated translations of up to 10% of the

Table 4: **Top-1 accuracy (%) on the CO3D dataset using k-Nearest Neighbors (k=1).** We compare pretrained vision foundation models, an image-based baseline, and our proposed video-based fine-tuning method. ExPLoRA and VESSA results are reported on the validation set using representations extracted from the backbone and evaluated via k-NN. We report confidence intervals to highlight the statistical significance of the improvements.

| Method | DINO-B [3] | DINOv2 [4] | TIPS [43] |
|---|---|---|---|
| ExPLoRA [28] + video | $83.64 \pm 0.84$ | $89.64 \pm 0.47$ | — |
| VESSA (ours) | $\mathbf{85.03} \pm 0.52$ | $\mathbf{91.85} \pm 0.56$ | $\mathbf{70.56} \pm 1.03$ |

Table 5: **Top-1 accuracy (%) on the MVImageNet dataset using k-Nearest Neighbors (k=1).** We compare pretrained vision foundation models, an image-based baseline, and our proposed video-based fine-tuning method. ExPLoRA and VESSA results are reported on the validation set using representations extracted from the backbone and evaluated via k-NN. We report confidence intervals to highlight the statistical significance of the improvements.

| Method | DINO-B [3] | DINOv2-B [4] | TIPS-B [43] |
|---|---|---|---|
| ExPLoRA [28] + video | $87.74 \pm 1.03$ | $\mathbf{96.15} \pm 0.87$ | — |
| VESSA (ours) | $\mathbf{92.51} \pm 1.11$ | $96.01 \pm 1.08$ | $\mathbf{80.54} \pm 1.71$ |

image dimensions, rotations up to 10 degrees, scaling variations up to 5%, brightness shifts of 0.1, and contrast adjustments in the range of 0.9 to 1.1. These transformations were carefully selected to approximate changes in camera perspective while avoiding the introduction of unrealistic artifacts. As shown in Table 6, these modifications did not yield significant performance improvements compared to the baseline using standard image augmentations, suggesting that the advantages observed with real videos may stem from cues beyond simple geometric or photometric variation.

**The data augmentation pipeline** consists of two global views and multiple local crops, each subjected to a specific set of transformations, as detailed below.

- **Global crops:** Two crops are sampled with scale ranges between $(0.4, 1.0)$. The transformations applied to these global crops are as follows:
  - **Transformation view 1:** horizontal flip (probability 0.5), color jitter (strength 0.8), grayscale conversion (probability 0.2), and Gaussian blur (probability 1.0).
  - **Transformation view 2:** horizontal flip (probability 0.5), color jitter (strength 0.8), grayscale conversion (probability 0.2), Gaussian blur (probability 0.1), and solarization (probability 0.2).
- **Local crops:** A set of $u$ local crops per image are sampled with a scale range $(0.05, 0.25)$ defined by the configuration and resized to $96 \times 96$ pixels. Each local crop undergoes the following transformations independently:
  - Color jitter with parameters (strength 0.8, brightness 0.4, contrast 0.4, saturation 0.2, hue 0.1).
  - Grayscale conversion (probability 0.2).
  - Gaussian blur (probability 0.5).

**Impact of using video-based inputs.** To highlight the differences between the view generation strategies employed by VESSA and those used in DINO, we present illustrative examples of view pairs from both approaches. It is important to note that in both VESSA and DINO, the same groups of transformations are applied independently to each view. To assess the impact of video-based inputs on representation learning, we analyze input variability by contrasting the frame selection strategy used in VESSA with the standard augmentation-based sampling in DINO and DINOv2. As shown in Figure 4, frame pairs selected by VESSA—based on a fixed temporal offset of $\delta = 5$ frames—exhibit substantially greater visual diversity than those generated through standard augmentations. In contrast, the views generated by DINO/DINOv2 tend to be more visually homogeneous. This enhanced

Table 6: Performance comparison using our method and images and transformations to simulate camera movement in images with DINO and DINOv2 on the CO3D dataset with $k = 1$.

| Method | DINO - B | DINOv2 - B |
|---|---|---|
| VESSA | **85.03** | **91.85** |
| Static-baseline | 80.31 | 81.60 |
| Static-baseline + Transf. simulate video | 80.60 | 81.49 |

variability introduced by real video frames is likely a key factor in the performance differences observed when training with video data, as opposed to relying solely on static image augmentations.

**Impact of catastrophic forgetting and cross-dataset generalization.** We also investigate the impact of domain-specific adaptation with VESSA on the original pretraining task. To this end, we evaluate the performance of DINO, DINOv2, and TIPS on ImageNet classification using a k-nearest neighbors (KNN) classifier, both in their pretrained form and after adaptation on CO3D and MVImageNet. As shown in Table 7, while the pretrained models achieve competitive accuracy on ImageNet, their performance drops drastically once adapted with VESSA on either domain. This result confirms the presence of catastrophic forgetting and underscores the infeasibility of applying the adapted models in a general-purpose setting. Nonetheless, this outcome aligns with the intended design of VESSA: the method is tailored to specialize visual foundation models for unsupervised adaptation in a target domain, where it delivers strong performance despite losing generality.

Table 7: Effect of catastrophic forgetting on ImageNet classification after VESSA adaptation.

| Model | DINO | DINOv2 | TIPS |
|---|---|---|---|
| Pretrained | **76.10** | **82.10** | **80.00** |
| VESSA (CO3D) | 15.46 | 17.15 | 18.10 |
| VESSA (MVImgnet) | 15.68 | 16.78 | 17.10 |

Moreover, we conducted an experiment in which training was performed using VESSA exclusively on the MVImageNet dataset, followed by evaluation on the held-out test set of the CO3D dataset. As shown in Table 8, this cross-dataset setting reveals a marked drop in performance—approximately 5 to 7 percentage points—when compared to the baseline results obtained by pretraining and evaluating on the same dataset. This performance degradation highlights the presence of catastrophic forgetting and limited generalization capabilities when the model is exposed to a distribution shift, even when trained on a diverse and temporally rich video corpus.

Table 8: Performance comparison between DINO and DINOv2 models using the pretrained base model, our proposed VESSA method, and the cross-dataset evaluation. The cross-dataset model was trained on MVImageNet and tested on CO3D to analyze forgetting behavior, demonstrating the degradation experienced when a model is trained on one dataset and evaluated on another. All experiments utilized the ViT-B architecture.

| Method | DINO | DINOv2 |
|---|---|---|
| Pretrained | 78.86 | 87.86 |
| Cross dataset | 74.40 | 80.36 |

**Training cost and computational efficiency.** We also analyze the computational requirements of adapting visual foundation models with VESSA to assess its practical feasibility. For the Co3D dataset ($20,412$ training pairs and $4,535$ test samples) using a ViT-Base backbone, the adaptation stage required only 1.97 hours of training, corresponding to 7.04 core-hours on a TPU v3-8, which consists of 4 TPUs, each with 2 cores (total of 8 cores). The process ran for 20 epochs, processing $407,040$ examples at a throughput of 135 images per second (16.88 images per second per core), with a total energy consumption of 13.86 kWh and an estimated carbon footprint of $5.55kg$ $CO_2$. Importantly, VESSA adapts a pre-trained model rather than training from scratch, and the minor additional overhead introduced by processing paired video frames (or local crops) is negligible due to

offline video decoding. This efficiency contrasts sharply with the full pretraining of models such as DINO or DINOv2, which require thousands of GPU-hours and result in $CO_2$ emissions on the order of tons.

