# OpenReview forum: "VESSA: Video-based objEct-centric Self-Supervised Adaptation for Visual Foundation Models"
_NeurIPS.cc/2025/Conference — NeurIPS 2025 poster_

### Official Review · Reviewer_5Abh · 2025-06-21

**Clarity:** 3
**Significance:** 3
**Originality:** 2
**Rating:** 5
**Confidence:** 4

**Summary:**

This paper proposes the VESSA framework, which uses object-centric videos to improve the generalization ability of vision foundation models in an unsupervised manner. By treating the temporally distinct frames in the same video as positive pairs during contrastive training, this method enables learning robust and transferable representations.

**Questions:**

Please refer to the 'Weaknesses' section above.

**Ethical Concerns:**

["NO or VERY MINOR ethics concerns only"]

**Final Justification:**

This work introduces an effective and efficient unsupervised domain adaptation method. The effectiveness is validated through extensive ablations on key module designs and results on different datasets/models. The efficiency is stated with training cost comparisons. Implementation details are provided for reproducibility. Catastrophic knowledge forgetting during transfer is a major limitation and is quantified. Nonetheless, this method provides valuable insights for adaptation research and practical scenarios.

**Limitations:**

Yes.

**Paper Formatting Concerns:**

None.

**Quality:**

3

**Strengths And Weaknesses:**

**Strengths**

**1. [Good presentation]** The writing is clear and easy to follow.

**2. [Well-illustrated figures]** The overview and pipeline figures are clear enough to show the workflow of the method.

**3. [Carefully-designed modules]** The method proposes several necessary modules and validates through a comprehensive ablation.
(i) Frame Selection module: In Table 1, using video as input consistently outperforms variants using image as input under the same setting. In Table 3, the authors also validates that using frame distance sampled from a random interval performs best.
(ii) Preprocessing and Augmentation module: The random augmentaion as well as the _local crops_ are fundamental for VESSA. Using _local crops_ results in a 1.34% improvement in Table 1.
(iii) The UWSD loss: The model can be more aware of hard samples. Using USWD loss provides a 0.95% improvement on MVImageNet.
(iv) Fine-tuning strategy: The staged unfreezing optimization strategy detailed in Section 3 provide insights in how to prevent representational drift when fine-tuning a pre-trained foundation model.

**4. [Thorough evaluations]** The experimental results show the model's superioity.
(i) In Table 3, VESSA outperforms other baselines at most cases.
(ii) Figure 3 demonstrates the effectiveness of VESSA using complicated samples based on nearest neighbor retrieval.

**Weaknesses**

**1. [Inadequate experiments]** More experimental results are necessary for varifying the efficiency and effectiveness of the method. (i) Ablations on the number of local crops: In Section 3, the paper mentioned that a set of _u_ pairs of local crops are processed by the student model. But it is unclear what this number is set and its influence on the final performance.
(ii) Training cost comparisons on video-based and image-based method. Although VESSA outperforms its static counterpart, this method requires paired input as well as _local crops_ during training. Comparisons on the training cost would be more intuitive to show the efficient adaptation progress.

**2. [Need further explorations]** Although the method has shown potential in classification task, it would be more convincing to show whether this framework is robust and generalizable to other tasks.
(i) In Figure 3, based on the retrieval results, VESSA focus more on **object**, which may benefit from the object-centric videos during adaptation. It would be more convincing to show whether such representation improvement can generalize to other **object-centric** tasks and more quantitative results are encouraged. Instance-level recognition experiments as validated in DINOv2[1] are recommended (but not necessary).
(ii) Since VESSA incorporates temporal dynamic information, comparisons on video classification tasks with foundation models are also encouraged.

[1] DINOv2: Learning Robust Visual Features without Supervision. Transactions on Machine Learning Research Journal, 2024.

---

> ### Author Rebuttal · Authors · 2025-07-31
>
> We sincerely thank the reviewer for the constructive and detailed feedback. We are glad that the reviewer found the paper well written, with clear figures, carefully designed modules, and thorough evaluations. Below, we address each of the concerns.
>
> > W1: [Inadequate experiments] More experimental results are necessary for varifying the efficiency and effectiveness of the method. (i) Ablations on the number of local crops: In Section 3, the paper mentioned that a set of u pairs of local crops are processed by the student model. But it is unclear what this number is set and its influence on the final performance. (ii) Training cost comparisons on video-based and image-based methods. Although VESSA outperforms its static counterpart, this method requires paired input as well as local crops during training. Comparisons on the training cost would be more intuitive to show the efficient adaptation progress.
>
> **(i) Ablations on the number of local crops**
>
> We appreciate the suggestion to further analyze the impact of the number of local crops. In our experiments, we adopted a fixed configuration following standard practices in state-of-the-art self-supervised methods (e.g., DINOv2), which has proven to be effective and stable across different datasets. Due to space constraints and in order to maintain focus on the primary contributions of our work, we did not include a variation of such parameters. However, we would like to remind the reviewer that we already include an experiment without local crops in Table 1 (row 2), which achieves an accuracy of 90.53, while the version using the default DINO number of local crops reaches 91.87. The improvement using local crops is significantly smaller than the improvements using video inputs (compare row 1 to row 8: 91.87 vs 88.54) or the improvements by adapting the training head before updating the backbone (compare row 1 to row 7: 91.87 vs 80.87). Therefore, we believe that variations in the number of local crops would not bring as significant gains as the other aspects we emphasize as main contributions in our work.
>
> **(ii) Training cost comparisons between video-based and image-based methods**
>
> We agree that reporting training costs is valuable to assess the efficiency of VESSA. Our method introduces a minimal computational overhead compared to its static counterpart. In the image-based method (VESSA variant that receives only images as input, Table 1 row 8), each training step processes two augmented views of the same image. In the video-based method (VESSA variant that receives video frames as input), each training step processes two distinct frames from the same video. In both cases, the model processes exactly two images per step, leading to nearly identical GPU memory usage and computation cost.
>
> The primary additional cost comes from video decoding, but this step is performed offline as a preprocessing stage, where all videos are decoded and frames are extracted at a fixed FPS before training. During adaptation, the training pipeline only samples pre-extracted frames, resulting in negligible additional cost and no significant impact on the overall throughput of model optimization.
>
> It is also important to note that VESSA is not trained from scratch. Instead, it builds upon the publicly available pre-trained weights of DINOv2.. For our unsupervised adaptation stage, we measured the actual training cost (details also provided in our response to Reviewer 3, jCuG). Using a ViT-Base backbone on the Co3D dataset (20,412 training pairs, 4,535 test samples), the adaptation stage required only 1.97 hours of training (7.04 core hours) on a TPU v3-8, which consists of 4 TPUs, each with 2 cores (total of 8 cores) , consuming 13.86 kWh of energy and resulting in a carbon footprint of just 5.55 kg CO2 for 20 epochs.
>
> This is negligible compared to the original DINOv2 training cost: according to their paper, reproducing DINOv2 alone requires 22,016 GPU-hours (9.7 MWh), emitting approximately 3.7 tons of CO2. The entire DINOv2 project is estimated to have a footprint between 0.5k and 1k tons of CO2 (around 200k GPU-days). In comparison, our adaptation stage emits a lot less CO2 than retraining DINOv2 from scratch.
>
> Given the substantial performance gains demonstrated in Tables 1 and 3, we believe this trade-off is favorable. We will include runtime metrics in the final version of the paper and provide scripts in the released code to facilitate further cost analysis by the community.
>
> > W2: [Need further explorations] Although the method has shown potential in classification task, it would be more convincing to show whether this framework is robust and generalizable to other tasks. (i) In Figure 3, based on the retrieval results, VESSA focus more on object, which may benefit from the object-centric videos during adaptation. It would be more convincing to show whether such representation improvement can generalize to other object-centric tasks and more quantitative results are encouraged. Instance-level recognition experiments as validated in DINOv2[1] are recommended (but not necessary). (ii) Since VESSA incorporates temporal dynamic information, comparisons on video classification tasks with foundation models are also encouraged.
>
> Our primary goal in this work is to develop an effective unsupervised adaptation method tailored to specific downstream tasks, rather than creating a universal model that performs optimally on a wide variety of tasks out of the box. This task-specific adaptation focus is highly relevant in practical applications, where efficient tuning to the target domain is often more valuable than large-scale pretraining alone. While we currently do not include instance-level recognition or video classification experiments, we acknowledge the reviewer’s comment that such experiments are not strictly necessary. We agree that, with appropriate adjustments, VESSA’s use of temporal dynamics and object-centric video information could be extended to these tasks. Extending our evaluations to these settings is part of our planned future work. We view these extensions as promising research directions that naturally follow the contributions of this paper.

---

> > ### Comment · Reviewer_5Abh · 2025-08-05
> >
> > Thanks for the detailed clarification. I have read all the rebuttal content and my concerns have beed addressed.

---

### Official Review · Reviewer_jCuG · 2025-06-28

**Clarity:** 3
**Significance:** 3
**Originality:** 3
**Rating:** 5
**Confidence:** 3

**Summary:**

This paper proposes VESSA, an video-based self-supervised fine-tuning method for visual foundation models (VFMs) to achieve domain adaptation without labeled data. The method leverages object-centric temporal consistency in videos by pairing frames of the same object at different timestamps within a self-distillation framework to adapt feature representations. To mitigate forgetting during fine-tuning, the authors adopt a progressive fine-tuning scheme: first training the prediction head, then gradually unfreezing backbone layers, combined with LoRA for parameter-efficient tuning. Additionally, they introduce an Uncertainty-Weighted Self-Distillation (UWSD) loss that emphasizes learning from hard samples by weighting the distillation loss based on teacher model uncertainty. Extensive experiments on three state-of-the-art VFMs (DINO, DINOv2, TIPS) and two large-scale multi-view video datasets (MVImageNet, CO3D) demonstrate consistent improvement over original pre-trained models and competitive baselines.

**Questions:**

Training Cost Clarification: Please provide detailed information about the training cost, including total training time, computational resource consumption (e.g., GPU/TPU hours), and differences across models or dataset scales, to better understand the method’s practical feasibility.

Data Dependency and Generalization: How does the method perform when access to dense multi-view object-centric videos is limited? Could it be adapted or generalized to settings with fewer or no multi-view frames?

Forgetting Mitigation: Have you explored additional strategies to mitigate forgetting during fine-tuning beyond partial layer unfreezing and LoRA? Quantitative forgetting metrics or further regularization techniques would strengthen the analysis.

**Ethical Concerns:**

["NO or VERY MINOR ethics concerns only"]

**Final Justification:**

My concerns have been addressed. I will keep the "Rating: 5: Accept".

**Limitations:**

yes, authors addressed the limitations and potential negative societal impact of their work

**Quality:**

3

**Strengths And Weaknesses:**

Strengths:
- Innovative Technical Contributions: The paper effectively utilizes object-level video temporal consistency and uncertainty-weighted self-distillation to improve unsupervised domain adaptation of VFMs.
- Comprehensive Experiments and detailed analysis: Evaluations on multiple backbone models and datasets corroborate the method’s general applicability and robustness.
- Potential for Real-world Use: The method requires no manual annotations and can adapt to new data distributions with unlabeled videos.

Weaknesses
- Training Cost Not Fully Quantified: The paper reports hardware used (TPU v3-8) but lacks detailed analysis of training time, total compute requirements, or energy consumption.
- Dependence on Multi-view Video Data: Reliance on videos depicting the same object from multiple viewpoints limits applicability where such data is scarce.
- Forgetting Issue: Although acknowledged, there is limited discussion or experimental analysis on how forgetting is quantitatively assessed or mitigated beyond partial layer unfreezing.
- Some Implementation Details Could Be Clearer: Additional details on hyper-parameter choices would improve reproducibility.

---

> ### Author Rebuttal · Authors · 2025-07-31
>
> We greatly appreciate the reviewer's thoughtful and detailed feedback. We are glad that our experiments are recognized as comprehensive, validating our method’s general applicability and robustness. Additionally, we appreciate the recognition of the method’s potential for real-world applications. The suggestions will be incorporated into the final version of the manuscript.
> Please find our detailed responses and clarifications below.
>
> > Q1 (W1): Training Cost Clarification: Please provide detailed information about the training cost, including total training time, computational resource consumption (e.g., GPU/TPU hours), and differences across models or dataset scales, to better understand the method’s practical feasibility.
>
> Thank you for raising this question. These details are indeed important and will be included in the final version of the paper. Below, we present an analysis from a single execution of VESSA in an adaptation process for the Co3D dataset (20,412 training pairs and 4,535 test samples) using the ViT-Base architecture.
> The adaptation required 1.97 hours of training, corresponding to 7.04 core-hours. The total energy consumption was approximately 13.86 kWh, which translates into a carbon footprint of 5.55 kg CO₂, assuming an average emission factor of 0.4 kg CO₂/kWh (typical for data centers). The process ran for 20 epochs, processing a total of 407,040 examples at a throughput of 135 images/second (or 16.88 images/second per core), with an average rate of 1.0485 steps/second. The experiment used 8 devices in parallel, each with a local batch size of 128 and a per-device batch size of 16. The ViT-Base model contained 111,472,896 trainable parameters (including Lora parameters) during adaptation.
> It is worth noting that, unlike DINO, which trains from scratch, VESSA adapts a pre-trained model. Even considering this distinction, when compared to the reported training cost of the original DINO model on ImageNet, as described in Emerging Properties in Self-Supervised Vision Transformers (Caron et al., 2021), the efficiency gains are clear. DINO’s full pretraining involved approximately 1,024 GPU-hours on V100 (equivalent to hundreds of kWh and a proportionally higher CO₂ footprint), while VESSA completes adaptation in less than two hours, with only a small fraction of the computational and environmental cost. This makes VESSA a computationally efficient and environmentally sustainable option for domain adaptation.
>
> > Q2 (W2): Data Dependency and Generalization: How does the method perform when access to dense multi-view object-centric videos is limited? Could it be adapted or generalized to settings with fewer or no multi-view frames?
>
> We would like to emphasize that the primary goal of our method is adaptation using object-centric videos, for which it performs well. In line with the idea of expanding the applicability of VESSA, there are several potential directions for future work. One promising avenue would be to incorporate an object tracking stage prior to applying VESSA. This tracking component would be responsible for following and extracting multiple views of the same object, enabling a multi-view VESSA setup even in unconstrained video environments. Such an extension could broaden the applicability of the method to scenarios with fewer or no dense multi-view frames.
>
> > Q3 (W3): Forgetting Mitigation: Have you explored additional strategies to mitigate forgetting during fine-tuning beyond partial layer unfreezing and LoRA? Quantitative forgetting metrics or further regularization techniques would strengthen the analysis.
>
> We acknowledge the concern regarding the potential impact of domain-specific adaptation on performance in broader, more general tasks. Our approach is deliberately designed to tailor the model to a specific target domain, where it consistently delivers strong results. Nonetheless, recognizing the relevance of this question, we conducted an additional evaluation to measure classification performance on the ImageNet dataset using a KNN classifier. In this assessment, we compared the visual foundation models under consideration (DINO, DINOv2, and TIPS) both in their original pre-trained state and after adaptation with VESSA on each of the target domains (CO3D and MVImageNet).
>
> | Model               | DINO  | DINOv2 | TIPS  |
> |---------------------|-------|--------|-------|
> | Pretrained          | **76.10** | **82.10**  | **80.0**  |
> | VESSA (CO3D)        | 15.46 | 17.15  | 18.10 |
> | VESSA (MVImgnet)    | 15.68 | 16.78  | 17.10 |
>
> The first row of the table reports the baseline performance of each pre-trained visual foundation model when evaluated on ImageNet. The second row presents the results on the same dataset after adapting the models with VESSA using the CO3D data, where a substantial decrease in accuracy can be observed. A similar reduction appears in the third row, corresponding to models adapted on the MVImageNet dataset. These findings highlight that the adapted models are not suitable for general-purpose use. However, it is important to emphasize that, in practical scenarios, the adaptation is intended for a specific target domain, where performance continues to improve and remains highly effective for that domain.
>
> > W4: Some Implementation Details Could Be Clearer: Additional details on hyper-parameter choices would improve reproducibility.
>
> We appreciate this observation and will include more comprehensive details on the hyperparameter choices in the final version of the manuscript, noting that all implementation details will also be made available in our GitHub repository. The VESSA learning technique was implemented on top of DINO, primarily to benefit from its stable training setup and the extensive empirical tuning described in the original DINO work, as well as the similar training environment it offers. Consequently, many update-related hyperparameters were inherited directly, including: global_crops_scale (0.14, 1.0), local_crops_scale (0.05, 0.25), student_temp (0.1), center_momentum (0.9), number_local_crops (10), warmup_teacher_temp (0.04), teacher_temp (0.07), learning rate (0.001), learning_rate_schedule ("compound"), factors ("constant * cosine_decay * linear_warmup"), warmup_steps (config.steps_per_epoch * 15), steps_per_cycle (total_steps), base_learning_rate (lr * batch_size / 1024.), and alpha (0.01).
> Some settings were modified to meet machine memory limitations and to incorporate VESSA-specific adjustments. These include: epochs (20), lora_rank (64), batch_size (128), gamma (UWSD) (1.0), and number_pairs_video (3). These modifications were chosen to align with the available computational resources while preserving the method’s effectiveness.
>
> We trust that these answers offer a more comprehensive view of VESSA’s training cost, data requirements and generalization behavior, as well as its handling of forgetting and implementation aspects. All points raised will be incorporated into the final version of the paper.

---

> > ### Comment · Reviewer_jCuG · 2025-08-06
> >
> > Thanks for the detailed reply. My concerns have been addressed. I will keep the "Rating: 5: Accept".

---

### Official Review · Reviewer_49Dk · 2025-07-01

**Clarity:** 3
**Significance:** 3
**Originality:** 2
**Rating:** 4
**Confidence:** 4

**Summary:**

This paper proposes to adapt Vision Foundation Models such as DINO or DINOv2 using video from multi-view object centric datasets. For this, they first formulate a frame selection criteria, followed by a multi-crop strategy similar to DINO and an uncertainty-weighted loss. The evaluations are conducted on these same multi-view object-centric datasets, CO3D and MVImageNet, and they show promising results. Also, the components of the method are thoroughly ablated.

**Questions:**

Overall, I believe the paper requires some revisions. As described in the weaknesses, some tables are a bit overcrowded, and should rather be labeled as ablation tables. The combination of different aspects of the method opens up a discussion if the proposed technical novelties of the paper are the most important for making the method work.
Furthermore, I wonder how this adaptation of DINO also improves its performance on other tasks/datasets? Does this approach also work for other self-supervised VFMs such as iBOT or I-JEPA? These are just suggestions, other VFMs are fine to try, too.

If the authors address my concerns in a convincing way, I’m considering to update my score.

**Ethical Concerns:**

["NO or VERY MINOR ethics concerns only"]

**Final Justification:**

After the rebuttal, the authors put in significant efforts to address my concerns. After the discussion, and the title/abstract updates, I raise my score to Borderline Accept.

**Limitations:**

Yes

**Quality:**

3

**Strengths And Weaknesses:**

Strengths:
- The overall idea of the paper is well motivated, sound and interesting
- The writing of the paper is mostly clear and easy to understand
- The ablations of the method is thorough and extensive, but presented maybe a bit confusingly
- The results in Table 3 show that the method can be effective across multiple DINO backbones

Weaknesses:
- Limited applicability: While the method is being advertised as a video-based adaptation mechanism for VFMs, it’s only designed to work on multi-view object centric datasets, i.e. it is not applicable to any in-the-wild video. With this in mind, I think the framing of the overall paper, including the title, should be narrowed to this use case
- The main VFM pre-training is not significantly changed, the model is trained with the standard loss with an added Uncertainty Weighting. This makes the technical contribution a bit thin.
- In Line 253, the authors cite their motivation for using temporal information to improve the VFM. While this is technically not false, I think describing at „using views of an object from a different angle“ would probably refine it.
- Table 1 is overfilled and a bit confusing. It’s hard to piece together what actually works and how doesn’t different numbers of unfrozen layers and the other components. Also, using local crops seems to bring the biggest increase in performance, a technique already used for the original DINO. This begs the question how relevant the proposed technical contributions are for the performance in relation.
- Tables 1 and 2 are both ablation tables and probably should be moved to a separate ablation section.
- It would be interesting to learn if the fine-tuning of DINO with multi-view datasets have improved its performance on benchmarks other than CO3D and MVImageNet
- Extending the approach to other VFM such as iBOT [1] or I-JEPA [2] would also be interesting

[1] Zhou, Jinghao, et al. "Image BERT Pre-training with Online Tokenizer." International Conference on Learning Representations.
[2] Assran, Mahmoud, et al. "Self-supervised learning from images with a joint-embedding predictive architecture." Proceedings of the IEEE/CVF Conference on Computer Vision and Pattern Recognition. 2023.

---

> ### Author Rebuttal · Authors · 2025-07-31
>
> We sincerely thank the reviewer for the thoughtful and constructive feedback. We are glad that the reviewer found the paper well-motivated, sound, and interesting, with clear writing and thorough ablations. We appreciate the suggestions for improving clarity, refining the scope and framing of the paper, and exploring additional evaluations. Below, we address each of the reviewer’s concerns in detail.
>
> > W1: Limited applicability: While the method is being advertised as a video-based adaptation mechanism for VFMs, it’s only designed to work on multi-view object centric datasets, i.e. it is not applicable to any in-the-wild video. With this in mind, I think the framing of the overall paper, including the title, should be narrowed to this use case
>
> We acknowledge that VESSA was specifically applied to object-centric videos in this work. We agree with the reviewer that clarifying this scope in the framing and title of the paper would improve its accuracy, and we are committed to making these changes. Despite this focus, the method already achieves significant improvements over the baseline on challenging multi-view datasets, demonstrating its practical value.
> We also recognize the importance of extending VESSA to handle in-the-wild videos with more complex dynamics, which we consider an important direction for future research. One potential avenue for this extension is the incorporation of an object tracking method prior to applying VESSA. Such a pre-processing step could help isolate objects in more cluttered scenes with multiple objects or significant camera motion. While we have not experimentally validated this yet, we believe this approach could allow VESSA to operate effectively in more realistic video scenarios, and we plan to explore it in future work.
>
> > W2: The main VFM pre-training is not significantly changed, the model is trained with the standard loss with an added Uncertainty Weighting. This makes the technical contribution a bit thin.
>
> It is important to emphasize that our contribution goes far beyond simply adding the Uncertanty-Weighted Self-Distillation (UWSD) loss. VESSA introduces a comprehensive self-supervised adaptation framework for vision foundation models (VFMs), specifically designed to leverage video information efficiently. To ensure stable adaptation, we propose a staged unfreezing approach, where we (i) initially freeze the backbone and train only the projection head, (ii) progressively unfreeze layers, and (iii) apply LoRA-based low-rank adaptation to early transformer layers, while fully fine-tuning later layers. This carefully designed strategy mitigates gradient instabilities and preserves valuable pretrained features. These combined elements form a lightweight yet effective adaptation method that consistently improves classification accuracy on domain-specific datasets (up to +10.54% in Table 3) while requiring minimal computational overhead. We therefore respectfully disagree with the characterization of the contribution as “thin”; rather, we believe the paper offers a significant and practically relevant step toward unsupervised adaptation of VFMs.
>
> > W3: In Line 253, the authors cite their motivation for using temporal information to improve the VFM. While this is technically not false, I think describing at „using views of an object from a different angle“ would probably refine it.
>
> We appreciate this observation. While we agree that the goal of our approach is to leverage different views of an object, it is important to clarify that the benefit comes specifically from using frames of the same physical object captured over time in a video, rather than simply using different objects of the same category viewed from different angles (e.g., a front view of one wooden chair and a side view of a different plastic chair). Using frames from the same video ensures that the model learns consistent, object-centric representations and focuses less on background information, as also highlighted by our qualitative results. To make this distinction clearer, we will refine the phrasing to “using different views of the object from the video” in the final version of the paper.
>
> > W4: Table 1 is overfilled and a bit confusing. It’s hard to piece together what actually works and how doesn’t different numbers of unfrozen layers and the other components. Also, using local crops seems to bring the biggest increase in performance, a technique already used for the original DINO. This begs the question how relevant the proposed technical contributions are for the performance in relation.
>
> For the final version, we will expand the results and improve the presentation of the tables to make the contributions of each component clearer. We would like to clarify that, contrary to the impression that local crops bring the largest performance gain, other components — particularly training the classification head and using video as input — have a much stronger impact on the results.
> For example:
> - Using images as input with local crops (Table 1, top section, last row) achieves 88.54% accuracy, while switching to video input **without** local crops (Table 1, top section, row 2) already improves performance to 90.53%, a much larger gain than local crops alone.
> - When combining video input with local crops and training the head, accuracy reaches 91.87%, whereas using the same video setup with local crops but without training the head (Table 1, top section, penultimate row) drops to 80.87%.
> This demonstrates that training the head and leveraging temporal information from videos are the dominant factors driving VESSA’s improvements, while local crops provide additional but comparatively smaller gains.
> The ablation table was designed to explicitly highlight these decisions and show why each proposed step (video-based adaptation, staged unfreezing, head training, local crops, and UWSD loss) is necessary. We will make these findings clearer in the final version to avoid misinterpretation.
>
> > W5: Tables 1 and 2 are both ablation tables and probably should be moved to a separate ablation section.
>
> We appreciate this suggestion. To improve the organization and clarity of the results presentation, we plan to create a dedicated ablation subsection in the final version of the paper. This subsection will group Tables 1 and 2 together and provide a more structured discussion of the ablation findings, making it easier for readers to follow the contribution of each component.
>
> > W6: It would be interesting to learn if the fine-tuning of DINO with multi-view datasets have improved its performance on benchmarks other than CO3D and MVImageNet
>
> We would like to emphasize that our focus is not on creating a universal model that performs well across all multi-view datasets. Instead, VESSA is designed for efficient, domain-specific adaptation of visual foundation models. To address the reviewer’s curiosity, we conducted an additional experiment where we adapted the visual foundation models DINO and DINOv2 to MVImageNet and evaluated them on CO3D, aiming to understand the behavior when the model is adapted to one dataset and tested on another.
>
> | Method         | DINO-B     | DINOv2-B   |
> |--------------------|------------|------------|
> | Pretrained         | **78.86**      | **87.86**  |
> | Cross dataset      | 74.40  | 80.36      |
>
> The results in the above table show a clear degradation in performance for DINO and DINOv2 in the cross-dataset scenario, indicating that adaptation is domain-specific and that improvements on one multi-view dataset do not automatically transfer to another.
> This behavior could be seen as a limitation if the goal were to adapt the model to multi-view datasets generally, however our motivation is to apply VESSA towards specific domains, as previously mentioned. This is motivated by real-world applications, where practitioners often need to adapt a model to a specific target dataset or domain efficiently, rather than aiming for a single model that generalizes equally well across all possible scenarios.
>
> > W7: Extending the approach to other VFM such as iBOT [1] or I-JEPA [2] would also be interesting
>
> | Dataset   | CO3D  |
> |-----------|-------|
> | Pretrained| 62.79 |
> | VESSA     | **63.70** |
>
> We appreciate the reviewer’s suggestion to evaluate other VFMs. We conducted additional experiments applying VESSA to iBOT on the CO3D dataset. using the publicly available ViT-base weights from the iBOT student network. (pre-trained on ImageNet-22K), the accuracy improved from 62.79% to 63.70% after adaptation with VESSA. Unfortunately, I-JEPA does not release ViT-Base weights, which prevented us from performing similar experiments for this model. We will include these results in the final version of the paper.

---

> > ### Comment · Reviewer_49Dk · 2025-08-05
> > **Rebuttal Response**
> >
> > I thank the authors for their response, many of my concerns are well addressed.
> >
> > However, taking together the concern about scope/applicability (W1) and the author response for the cross-dataset evaluation (W6), I think it is crucial the authors reframe the scope and framing of their paper to be more narrow. Currently, especially the title and also the abstract suggest a more general video approach, i.e. "we propose to improve a VFM with any  video". This should be narrowed down by specifically using the term "multi-view object-centric videos" in the title and the abstract, to make it clear that this is the purpose of this paper.
> >
> > With this modification, I would lean towards acceptance of the paper. I'm curious to hear specific title and abstract modification suggestions from the authors.

---

> > > ### Author Response · Authors · 2025-08-06
> > >
> > > We thank the reviewer for the consideration and for the constructive comments. We agree with refining the scope, and we will revise the title, abstract, and any relevant statements in the text, according to the reviewer’s suggestion.
> > >
> > > We are pleased to adjust the title to: **“VESSA: Video-based objEct-centric Self-Supervised Adaptation for visual foundation models”**. We believe that this will better align the framing of the paper with its intended scope.
> > >
> > > We have also revised the abstract to follow the reviewer’s suggestion. Below, we present the new abstract with the **modified sentences highlighted**, and the **previous version** of the sentence is presented in **square brackets**.
> > >
> > >
> > > **Abstract (VESSA)**
> > >
> > > Foundation models have advanced computer vision by enabling strong performance across diverse tasks through large-scale pretraining and supervised fine-tuning. However, they may underperform in domains with distribution shifts and scarce labels, where supervised fine-tuning may be infeasible. While continued self-supervised learning for model adaptation is common for generative language models, this strategy has not proven effective for vision-centric encoder models. To address this challenge, we introduce a novel formulation of self-supervised fine-tuning for vision foundation models, where the model is adapted to a new domain without requiring annotations, leveraging only **short multi-view object-centric videos. [short object-centric videos]**. Our method is referred to as **VESSA: Video-based objEct-centric Self-Supervised Adaptation for visual foundation models [Video-based Efficient Self-Supervised Adaptation]**. VESSA’s training technique is based on a self-distillation paradigm, where it is critical to carefully tune prediction heads and deploy parameter-efficient adaptation techniques – otherwise, the model may quickly forget its pretrained knowledge and reach a degraded state. VESSA benefits significantly from **multi-view object observations [object observations]** sourced from different frames in **an object-centric video [a video]**, efficiently learning robustness to varied capture conditions, without the need of annotations. Through comprehensive experiments with 3 vision foundation models on 2 datasets, VESSA demonstrates consistent improvements in downstream classification tasks, compared to the base models and previous adaptation methods. Code and models will be released.
> > >
> > >
> > >
> > > We sincerely thank the reviewer for the thoughtful and constructive feedback provided. We greatly value these insights, which have been instrumental in improving the clarity and focus of our work. We would also like to reiterate our full availability to address any additional questions, comments, or suggestions that may arise.

---

### Official Review · Reviewer_EBTZ · 2025-07-02

**Clarity:** 4
**Significance:** 3
**Originality:** 3
**Rating:** 4
**Confidence:** 3

**Summary:**

This paper introduces a method for adapting pretrained VFMs to new domains using unlabeled object-centric videos. The approach employs self-distillation with temporally separated frame pairs as positive samples, combined with LoRA and a careful training schedule that includes staged unfreezing and uncertainty-weighted loss. Experiments on MVImageNet and CO3D datasets with DINO, DINOv2, and TIPS models demonstrate consistent improvements over pretrained baselines and existing adaptation methods.

**Questions:**

1. The experiments are conducted on datasets that feature largely object-centric videos. Could you elaborate on the expected performance and potential challenges of applying VESSA to more general, "in-the-wild" videos that may contain significant background clutter, camera motion, or multiple objects?

Please also check the "weakness" section.
I am willing to change my score if my concerns are addressed.

**Ethical Concerns:**

["NO or VERY MINOR ethics concerns only"]

**Final Justification:**

I thank the authors for their response to the review comments and the complementary experiments they have conducted. I have also examined the feedback from other reviewers and the authors' corresponding responses. I appreciate the authors for their transparency regarding the quantitative performance drop from catastrophic forgetting, which represents a limitation of the proposed method, as well as their suggestion for processing "in-the-wild" video data.

Taking these considerations into account, I raise the score for "originality" while maintaining the overall evaluation score unchanged.

**Limitations:**

Yes

**Quality:**

3

**Strengths And Weaknesses:**

Strengths:
1. The method is well-motivated, with a clear and intuitive approach to adapting foundation models using video data.
2. The training is lightweight by utilizing the limited unfrozen network layers and the adoption of LoRA.
3. The paper is supported by comprehensive experiments and ablation studies that validate the effectiveness of the proposed components and design choices.


Weaknesses:
1. It is unclear whether key implementation details from the original DINO framework, such as temperature setting and teacher centering, were utilized. Clarifying this and the specific hyperparameters would help disambiguate the performance gains attributable to the proposed methods (e.g. UWSD) versus standard training techniques
2. While the paper acknowledges catastrophic forgetting as a limitation, the extent to which it affects the model's performance on its original pretraining tasks is not explored. A preliminary analysis of this would provide a more complete picture of the trade-offs involved.

---

> ### Author Rebuttal · Authors · 2025-07-31
>
> We sincerely appreciate the reviewer's insightful feedback. Several strengths of our method were highlighted, including its motivation, clear/intuitive approach and its lightweight training. The experiments are regarded as comprehensive,validating the method’s effectiveness. We have carefully considered all the feedback and provided comprehensive responses below. All the suggested points and clarifications will be incorporated into the final version of the manuscript.
>
> > W1: It is unclear whether key implementation details from the original DINO framework, such as temperature setting and teacher centering, were utilized. Clarifying this and the specific hyperparameters would help disambiguate the performance gains attributable to the proposed methods (e.g. UWSD) versus standard training techniques
>
> The implementation of the VESSA update component was based on DINO for training stability reasons, given the extensive empirical search reported in the DINO work and the similar environment it provides. In particular, the temperature setting and teacher centering were utilized, exactly following the DINO setup. To make it absolutely clear, we list the hyperparameters in detail here: global_crops_scale (0.14, 1.0), local_crops_scale (0.05, 0.25), student_temp (0.1), center_momentum (0.9), number_local_crops (u = 10), warmup_teacher_temp (0.04), teacher_temp (0.07), learning rate (0.001), learning_rate_schedule ('compound'), factors ('constant * cosine_decay * linear_warmup'), warmup_steps (config.steps_per_epoch * 15), steps_per_cycle (total_steps), base_learning_rate (lr * batch_size / 1024.), and alpha (0.01).
> Other key parameters were adapted due to machine memory constraints and hyperparameters specific to VESSA. These are the key adaptations: epochs (20), lora_rank (64), batch_size (128), gamma (UWSD) (1.0), and number_pairs_video (3).
> These changes were made to ensure compatibility with the available computational resources while maintaining the effectiveness of the method. All the details will be made available on GitHub for transparency and reproducibility.
>
> > W2: While the paper acknowledges catastrophic forgetting as a limitation, the extent to which it affects the model's performance on its original pretraining tasks is not explored.
>
> We understand the concern regarding the impact of our domain-specific adaptation on more general tasks. We would like to emphasize that our goal is to adapt the model to a specific domain and apply it solely within that target domain, where it performs very effectively for this purpose. However, acknowledging the relevance of the question, we conducted an experiment to evaluate the classification results using KNN on the ImageNet dataset. In this experiment, we compared the performance of the visual foundation models in question (DINO, DINOv2, and TIPS) both pre-trained and after adaptation with VESSA on each of the domains (CO3D, MVImageNet).
>
> | Model               | DINO  | DINOv2 | TIPS  |
> |---------------------|-------|--------|-------|
> | Pretrained          | **76.10** | **82.10**  | **80.0**  |
> | VESSA (CO3D)        | 15.46 | 17.15  | 18.10 |
> | VESSA (MVImgnet)    | 15.68 | 16.78  | 17.10 |
>
> The first row of the table presents the results for each pre-trained VFM validated on ImageNet. The second row shows the validation results on the same ImageNet dataset after adaptation with VESSA on the CO3D data. It is evident that there is a drastic drop in performance. The same drop is observed in the third row, where the models are adapted on the MVImageNet dataset. This demonstrates the infeasibility of using the model in a general manner. Nonetheless, we reiterate that our goal is to adapt the model to a specific target domain without supervision, achieving excellent results in this setup.
>
> > Q1: The experiments are conducted on datasets that feature largely object-centric videos. Could you elaborate on the expected performance and potential challenges of applying VESSA to more general, "in-the-wild" videos that may contain significant background clutter, camera motion, or multiple objects?
>
> Thank you very much for the suggestion on ways to expand the application of VESSA. Some of these ideas are being considered for future work that will explore this direction. One potential solution to address the challenges posed by background clutter, dynamic camera motion, and multiple objects is the incorporation of an object tracking method prior to the application of VESSA. This approach, while relying on the quality of the tracking method to ensure effective object localization, would allow VESSA to extend its capabilities to handle scenes with multiple objects and significant camera motion. In this case, this expansion would enable VESSA to perform well even in more chaotic real-world video environments, albeit with some dependence on the precision of the object tracking method employed.
> We hope these responses provide a clearer understanding of the hyperparameters, the impact of adaptation on the visual foundation models, and a potential expansion in the application of VESSA. We greatly appreciate the feedback and look forward to further enhancing our work.

---

> > ### Comment · Reviewer_EBTZ · 2025-08-07
> >
> > I appreciate the authors' response. As detailed in "final justification", I raise the score for "originality" while keeping the overall score unchanged.

---

### Comment · Area_Chair_dJ2N · 2025-08-04
**Reviewer Ack Reminder from AC**

Hi Reviewers,

As the discussion deadline is approaching, if you haven’t done so already, could you please take a moment to acknowledge the rebuttal, revise your score if your opinion has changed, and post any follow-up comments or questions you may have?

Thanks for your time and contributions to the review process.

Best,
AC

---

### Note · Authors · 2025-08-15

We would like to thank all reviewers for their constructive and thoughtful feedback, as well as for the discussion during the review process. From the very beginning, we took every comment seriously and worked to address each concern in detail during the rebuttal. Our goal was to clarify technical aspects, strengthen the presentation, provide additional evidence, and ensure the novelty and relevance of the contributions of VESSA.

We are particularly encouraged by the positive developments that emerged after our responses. For instance:

* 49Dk initially expressed concerns about the scope and framing of VESSA’s applicability. After our detailed clarifications and proposed refinements to the title and abstract, the reviewer acknowledged the improved focus and indicated a willingness to lean toward acceptance.
* jCuG raised important questions about training costs, forgetting mitigation, and generalization related to VESSA. We provided comprehensive empirical data on computational efficiency, quantitative analysis of forgetting, and additional experiments on cross-dataset adaptation. These responses led the reviewer to confirm the “Accept” rating and acknowledge that their concerns were fully addressed.
* 5Abh requested further ablations and exploration of additional tasks. We explained our design decisions regarding local crop numbers and training costs in VESSA, and discussed planned future work on broader evaluations. The reviewer expressed satisfaction with our clarifications and confirmed that their concerns were resolved.
* EBTZ recognized the strong motivation and clear writing of VESSA, and appreciated our thorough ablations and promising results. We also took their suggestions seriously, clarifying technical contributions and improving presentation details such as table organization and scope framing, which contributed to a more precise and focused manuscript. The reviewer raised their originality score, showing clearer recognition of VESSA’s novel contributions.

These examples illustrate that the rebuttal phase was instrumental in clarifying misunderstandings, resolving questions, and demonstrating the robustness and significance of VESSA. We believe this process has significantly strengthened the manuscript and the contribution VESSA makes to the field.

We deeply appreciate the reviewers’ time, effort, and openness to discussion, and we hope responses reflect the seriousness and dedication with which we approached this review process.

---

### Decision · Program_Chairs · 2025-09-17

**Decision:**

Accept (poster)

**Comment:**

The paper proposes VESSA, a method for adapting vision foundation models to new domains using unlabeled object-centric videos. Reviewers found the approach well-motivated and clearly presented, with strong experimental validation and ablation studies. Key contributions include leveraging temporal consistency in videos, staged unfreezing with LoRA for efficient fine-tuning, and an uncertainty-weighted self-distillation loss. The method demonstrates consistent improvements across multiple backbones and datasets, while maintaining efficiency compared to training from scratch. Reviewers also appreciated the comprehensive rebuttal, which clarified implementation details, quantified training costs, analyzed forgetting effects, and refined the scope of the paper by explicitly framing it as applicable to multi-view object-centric videos.

Some weaknesses remain, though they are not deal-breaking. Reviewers noted that catastrophic forgetting is significant when models are applied back to general-purpose tasks, and that the method’s reliance on multi-view object-centric video data limits broader applicability. While suggestions for extending to in-the-wild video and additional tasks were raised, reviewers agreed that these fall under future work rather than immediate requirements. Overall, the paper makes a clear and meaningful contribution to efficient adaptation of vision foundation models, and the reviewers converged on acceptance after rebuttal and discussion.